# Phenotypic ranking experiments in identifying breeding objective traits of smallholder farmers in northwestern Ethiopia

**Oumer Sheriff** [1,2,3]*, **Kefyalew Alemayehu**[2,3], **Aynalem Haile**[4]

**1** Department of Animal Science, Assosa University, Assosa, Ethiopia, **2** Department of Animal Production and Technology, Bahir Dar University, Bahir Dar, Ethiopia, **3** Biotechnology Research Institute, Bahir Dar University, Bahir Dar, Ethiopia, **4** Resilient Agricultural Livelihood Systems Program (RALSP), International Center for Agricultural Research in the Dry Areas (ICARDA), Addis Ababa, Ethiopia

\* soumer74@yahoo.com

**Data Availability Statement:** All relevant data are within the paper and its Supporting Information files.

## Abstract

We executed two live animal ranking experiments, own-flock and group-animal ranking, to identify the breeding objectives of Arab and Oromo goat keepers in northwestern Ethiopia as a preliminary step towards designing sustainable breeding programs for two goat populations. In the own-flock ranking experiment, a total of 147 households, out of which 46 were Arab and 101 were Oromo goat keepers that live in semi-arid and sub-humid agroecologies respectively, were visited at their homesteads and were asked to choose their first best, second best, third best and the most inferior does from their own flock. The reasons of ranking and life history of the does (age, previous production and reproduction information) were inquired and recorded; live body weight and some linear body measurements were taken. In the group-animal ranking experiment, 12 breeding does and 12 breeding bucks from Arab goats and the same number of animals from Oromo goats were randomly selected. Life history of selected does and bucks (age, birth type, libido and temperament) were inquired from the owners. The selected animals were randomly grouped into four in Arab goats (three animals per group) and the same was applied in Oromo goats. Twelve farmers for Arab goats and the same number of farmers for Oromo goats who have not known the experimental animals were invited to do the ranking. Each person ranked the three animals in each group as 1st, 2nd and 3rd, giving reasons of ranking. After a first round of ranking, s/ he was then provided with the history of each individual animal and asked whether s/he would consider re-ranking them. This procedure was continued eight times until a person covered all groups of does and bucks. It was found out that in own-flock ranking experiment, keepers focus on productive, reproductive and behavioral traits (such as body size, mothering ability, twinning rate, kidding interval and temperament) while in group-animal ranking experiment, there was a general tendency to focus on observable physical traits like coat color, body size and body conformation. Simultaneous use of both own-flock and group-animal ranking experiments is advisable to identify breeding objective traits in production systems where record keeping is absent.

**Funding:** The research is funded by Biotechnology research institute of Bahir Dar University 1/4449/ 1.11.10 (BRI-BDU). The recipient of the fund is the corresponding author (Oumer Sheriff). However, the funders had no role in study design, data collection and analysis, decision to publish, or preparation of the manuscript.

**Competing interests:** The authors have no competing interests.

# Introduction

There are about 36.81 million goat populations in Ethiopia, of which 99.97% are local breeds [1]. Most of them are found in large flocks in arid and semi-arid lowlands while very small flock sizes are widely distributed in the highlands [2]. Goats play an important role in the smallholders' farming systems, for instance, they provide tangible (cash, milk, meat, fiber and manure) and intangible benefits (prestige, saving, insurance, cultural and ceremonial purposes) [3].

Given the presence of large number of goats and their diverse functions, the productivity of this valuable genetic resource is generally low. For instance, in the years (1999–2008), the average carcass weight produced from a yearling goat was only 8 kg; one of the lowest compared to the world average (12 kg) [4]. Likewise, the dressing percentage (DP) at one year of age is also very low (42–45%) [5]. The causes for poor performance of indigenous goats could be attributed to various interrelated factors [6, 7]. Among them, lack of suitable breeding programs is an important constraint.

Identifying the smallholder farmers' breeding objective traits is crucial to design appropriate breeding programs [8, 9]. Four different methods have been implemented to identify the breeding objective traits; for sheep and goats in Ethiopia. These include semi-structured questioner, choice card experiment, group discussion and ranking of live animals. While the first three were often used by many scholars [8, 10–14], the last method was brought forth by [10]. It has two forms: ranking of own animals with known history and ranking of animals with unknown history. One can use combination of the methods to determine the breeding objective traits for a given breed. Detailed descriptions of the methods are given elsewhere [10, 15].

Benishangul Gumuz region, our study area, is found in the northwestern lowlands of Ethiopia. The region is among the major goat production areas in the country, where Arab and Oromo goat populations and their crosses with other indigenous goats are widely distributed [16, 17]. The official census recorded 440,719 goats in 2015 [18] which are mostly produced in small familial units (on average eleven goats/household) for sale, own consumption, saving and cultural importance [13]. It can be said that the goat populations, including the study area, are untapped resources with very little research efforts. The breeding objective traits of the local goat keepers were not empirically identified and defined. Farmers simply select animals based on morphological features and production characteristics. This as it is, the local goat populations are noted for their low productivity [19]. The present study was therefore aimed at identifying breeding objective traits of Arab and Oromo goat keepers in northwestern Ethiopia, using own-flock and group-animal ranking approaches, as a preliminary step towards designing sustainable breeding.

# Materials and methods

## Ethics approval and consent to participate

The current study and the proposed parent study were approved by Bahir Dar University College of Agriculture and Environmental Studies (BDUCAES) and Bahir Dar University Biotechnology Research Institute (BDUBRI). Following endorsement by the BDUCAES and BDUBRI, Assosa University (AsU) was informed about the objectives of the study through a support letter (Ref. 1/2241/134 dated back to November 13, 2017) from BDUCAES. After reviewing the proposal, AsU wrote a permission and support letter to agricultural and rural development offices of Bambasi and Homosha districts. Then, the corresponding author of this paper and four development agents from the two districts selected the goat owners for the present study. Finally, the goat owners were informed about the research and asked for verbal

consent to confirm us to take the morphometric measurements on the selected goats. Four independent peasant association administrators acted as witnesses for voluntary informed decision making of the goat owners.

## Description of the study area

Detailed descriptions of the study areas and the goat populations found in the study areas were given elsewhere [13, 16]. In brief, the study was conducted in Bambasi and Homosha districts of Benishangul Gumuz region, northwestern Ethiopia. The districts were purposively selected to represent two different agroecologies, farming systems and goat populations. In each district, two peasant associations (PAs)—the lowest administrative units in Ethiopia, (Tumet and Sherkole from Homosha district and Bambasi 02 and Mutsa 01 from Bambasi district) were selected based on goat population size, presence of communal grazing areas, relative significance of goats to the livelihood of the communities, access to market and road. The number of sampled households was determined following [13]. Accordingly, the calculated number of households were 25 (Sherkole), 21 (Tumet), 42 (Bambasi 02) and 59 (Mutsa 01). This makes the total number of households covered in this study to be 147 (i.e., 46 from Homosha district and 101 from Bambasi district). Finally, households who owned at least four adult goats with a minimum of one year experience in goat husbandry and willing to participate in community-based breeding programs were identified. The list was prepared in each selected PA with the help of development agents. Respondents were selected from the prepared list using systematic random sampling technique until the calculated sample size of each PA was maintained.

Communities in Bambasi are mainly sedentary agriculturalists who keep Oromo goats–named after the Oromo community. These goats are meat type and are adapted to sub-humid agroecology. Maize, sorghum, finger millet, teff, haricot bean and sesame are among the crops produced in the area. Homosha is semi-arid area characterized by limited crop production due to poor soil fertility and unreliable rainfall. The Arab goats–named after the Arab/Berta community–predominate in the area.

## Own-flock ranking experiments

The data for this study were collected from January to February 2019. During the data collection, goat keepers were visited early in the morning at their homestead before their goats were let out for grazing. The goat keepers were asked to choose their 1st, 2nd and 3rd best and the most inferior does among the breeding does in their flock. Reasons for the ranking and life history of the ranked does (age, number of kidding, twinning ability, number of kids born per kidding and number of kids weaned) were inquired and recorded. As there are no records kept by the goat keepers in the study areas, family members who had participated in the ranking exercises were reminding each other about the history of their animals.

Body weight (BW) and linear body measurements (LBMs) such as body length (BL), chest girth (CG), wither height (WH) and rump height (RH) were taken from each animal as described by FAO (2012). BW is the fasted live body weight (in kg); BL is the horizontal distance (in cm) from the point of shoulder to the pin bone; CG is the circumference of the body (in cm) immediately behind the shoulder blades and perpendicular to the body axis; WH is the vertical height (in cm) from the bottom of the front foot to the highest point of the shoulder and RH is the vertical height from the bottom of the back foot to the highest point of the rump.

BW (kg) was recorded using suspended spring balance with 50 kg capacity and a precision of 200 g. Weighing sacks were used to lift goats during the BW measurements. The height measurements (cm) were taken using a graduated measuring stick while the length, width and

circumference measurements (cm) were measured with plastic measuring tape (1.50 m long with the precision of 2 cm). All measurements were taken after restraining and holding the goats in their natural position and before they were released for grazing to avoid the effect of feeding and watering on the goats' size and conformation [20].

## Group-animal ranking experiments

In these experiments, twelve breeding does and twelve breeding bucks from the Arab goats and the same number of does and bucks from the Oromo goats were randomly selected and marked. To avoid repeated measurements, the does were chosen from the own-flock ranking experiments covering all ranks. The information previously obtained from the owner (age, number of kidding, twinning ability, number of kids born per kidding and number of kids weaned) and the phenotypic measurements recorded during the own-flock experiments were used as life history for each selected doe. Similarly, the life history of the selected bucks (age, birth type, libido and temperament) and body weight measurements were inquired and recorded.

The selected animals were brought to a central place in each district and randomly assigned into groups. Animals of same sex were randomly assigned to four groups in Arab goats (three animals each) and the same was applied in Oromo goats. Twelve farmers for Arab goats and 12 farmers for Oromo goats, who have not known the selected animals, were then invited to rank the animals. Each farmer was inquired to rank the three animals in each group as 1$^{st}$, 2$^{nd}$ and 3$^{rd}$, and the reasons of ranking. The farmers were then provided with the history of each individual animal and asked whether they would re-rank the animals or not. This procedure was continued eight times until a farmer covered all groups of the animals.

## Data management and analysis

The data collected from the study area were arranged, coded and managed in Microsoft-Excel spread sheet for further analysis. Since the responses of the goat keepers for both own-flock and group-animal ranking experiments were open ended, reasons for ranking were first checked one by one to determine the trait levels and then coded. Based on the nature of data, different types of statistical analyses were used. The statistical software R [21] was used to analyze the data from the own-flock and group-animal ranking experiments. The frequency and proportion of breeding doe and buck traits preferred by the goat keepers in both experiments and rank proportions before and after provision of life history information in group-animal ranking experiment were analyzed by the 'gmodels package' of R, version 4.0.3 using the 'CrossTable function'. Similarly, the Mean ± SE values for dentition and some production and reproduction traits were analyzed using 'LSM (least squares mean) package version 3.5.2' of R [21] fitting the rank as fixed effects in the model.

## Results and discussions

### Doe traits in own-flock ranking experiments

The lists of preferred doe traits from the own-flock ranking experiments are summarized in Table 1. Although no organized breeding program is in place in the study area, goat keepers, however, select breeding does based on own memory and various attributes of the animal. Application of similar selection strategy for breeding does was also reported in Ethiopia [6, 12, 13] and elsewhere in Africa [22–24]. Mothering ability, kid growth, body size, twinning rate, coat color and body conformation were found to be the most important doe traits, in that order, influencing keepers' preference in Arab. They accounted for 56.05% of the total

**Table 1. List of doe traits in own-flock ranking experiment.**

| Traits | | Arab | | Oromo | |
|---|---|---|---|---|---|
| | | **Freq** | **%** | **Freq** | **%** |
| 1. | Body size | 29 | 9.24 | 44 | 8.43 |
| 2. | Kid growth | 33 | 10.51 | 72 | 13.79 |
| 3. | Kid size at birth | 20 | 6.37 | 71 | 13.60 |
| 4. | Mothering ability | 38 | 12.10 | 74 | 14.18 |
| 5. | Twinning rate | 29 | 9.24 | 77 | 14.75 |
| 6. | Kidding interval | 22 | 7.01 | 25 | 4.79 |
| 7. | Coat color | 24 | 7.64 | 39 | 7.47 |
| 8. | Body condition | 10 | 3.18 | 19 | 3.64 |
| 9. | Drought tolerance | 16 | 5.10 | - | - |
| 10. | Body conformation | 23 | 7.32 | 21 | 4.02 |
| 11. | Body length | 9 | 2.87 | 6 | 1.15 |
| 12. | Temperament | 12 | 3.82 | 10 | 1.92 |
| 13. | Sex of kid | 11 | 3.50 | 12 | 2.30 |
| 14. | Age at puberty | 11 | 3.50 | 20 | 3.83 |
| 15. | Pedigree | 14 | 4.46 | 22 | 4.21 |
| 16. | Foraging ability | 4 | 1.27 | 10 | 1.92 |
| 17. | Body width | 9 | 2.87 | - | - |
| | Sum | 314 | | 522 | |

proportions of mentioned traits. On the other hand, twinning rate, mothering ability, kid growth, kid size at birth and body size, in that order of importance, together contributed 64.75% of the total proportions of doe traits mentioned by Oromo goat keepers. Other important traits include kidding interval (7.01%), kid size at birth (6.37%) and drought tolerance (5.10%) in Arab and coat color (7.47%), kidding interval (4.79%) and pedigree (4.21%) in Oromo. In general, the breeding objective traits preferred by the goat keepers reflected what traits of Arab and Oromo does were appreciated by owners.

In the present study, we observed that both Arab and Oromo goat keepers generally focus on kid quality (such as kid growth and kid size at birth) and related reproductive traits (like twinning rate and mothering ability). However, there was noticeable difference in preference for some of these and other traits between the two goat keepers. For instance, in Arab goat keepers, drought tolerance and body width were mentioned as important traits but these traits were not mentioned at all by the Oromo goat keepers. This result is clearly associated with agro-ecology and breeding objective of the breeders. Arab goat keepers, who dwell in the semi-arid areas, were opted for does with better drought tolerance due to the harsher environment, in terms of feed and water shortage and prevalence of moisture stress in most parts of the year. They also associated wide bodied does with higher twinning rate, better mothering ability and high carcass yield. Similar findings on preference of drought tolerance and body width for goats and sheep in comparable environments were also reported in Ethiopia [6, 10, 11, 25].

The preference of big body size and fast kid growth as important traits in both study areas are expected when the main purpose of keeping goats is for cash income. In most of the time, goats with big body size have high market demand and fast growing goats reach market weight sooner. Oromo goat keepers mentioned kid size at birth quite frequently (13.60%) than Arab goat keepers. This may be due to the reason that Oromo does have significantly bigger body sizes than Arab does [16] so that their kids might be bigger at birth. In relation to this, [26] and [27] elucidated that maternal body size positively influences the weight of their progeny at

**Table 2. Mean ± SE values of traits in different rank groups of does from own-flock ranking experiment.**

| Goat population | Traits | p | Overall mean | Ranks | | | |
|---|---|---|---|---|---|---|---|
| | | | | 1 | 2 | 3 | Inferior |
| Arab | Dentition | NS | 3.24 ± 0.13 | 3.41 ± 0.14 | 3.37 ± 0.11 | 3.20 ± 0.14 | 2.98 ± 0.13 |
| | BW, kg | *** | 30.98 ± 0.37 | 31.94 ± 0.37[a] | 31.53 ± 0.28[ab] | 30.74 ± 0.40[c] | 29.72 ± 0.44[d] |
| | NK | *** | 2.99 ± 0.11 | 3.30 ± 0.13[a] | 3.07 ± 0.10[b] | 2.85 ± 0.11[c] | 2.72 ± 0.10[cd] |
| | Twinning | *** | 1.46 ± 0.07 | 1.72 ± 0.07[a] | 1.54 ± 0.07[b] | 1.24 ± 0.06[cd] | 1.33 ± 0.07[c] |
| | NKB | *** | 3.86 ± 0.18 | 5.00 ± 0.22[a] | 4.02 ± 0.20[b] | 3.26 ± 0.15[c] | 3.17 ± 0.16[cd] |
| | NKW | *** | 3.40 ± 0.16 | 4.48 ± 0.18[a] | 3.54 ± 0.16[b] | 2.78 ± 0.12[c] | 2.78 ± 0.16[c] |
| Oromo | Dentition | *** | 3.13 ± 0.15 | 3.53 ± 0.09[a] | 2.78 ± 0.07[d] | 3.07 ± 0.08[c] | 3.12 ± 0.09[b] |
| | BW, kg | *** | 32.15 ± 0.67 | 34.13 ± 0.31[a] | 31.21 ± 0.30[b] | 31.92 ± 0.37[b] | 31.47 ± 0.36[b] |
| | NK | *** | 3.22 ± 0.20 | 3.80 ± 0.11[a] | 2.88 ± 0.08[d] | 3.15 ± 0.11[b] | 3.03 ± 0.09[c] |
| | Twinning | *** | 1.29 ± 0.06 | 1.42 ± 0.05[a] | 1.36 ± 0.05[b] | 1.24 ± 0.04[c] | 1.15 ± 0.04[d] |
| | NKB | *** | 3.76 ± 0.37 | 4.84 ± 0.18[a] | 3.46 ± 0.12[b] | 3.52 ± 0.13[b] | 3.22 ± 0.11[c] |
| | NKW | *** | 3.32 ± 0.40 | 4.50 ± 0.15[a] | 3.00 ± 0.09[c] | 3.08 ± 0.10[b] | 2.71 ± 0.09[d] |

Row means within each goat population with different superscript letter are statistically different

***p ≤ 0.001; NS, non-significant; BW, body weight; NK, number of kidding; NKB, number of kids born/doe/kidding; and NKW, number of kids weaned/doe.

birth. Similarly, a relatively higher twinning rate (14.75%) as the preferred trait of Oromo goat keepers might be due to the availability of adequate feed throughout the year that can support many animals compared to the dry semi-arid area of Arab goat keepers.

None of the goat keepers in this study reported the use of goat milk. According to keepers, does with high milk production were considered as good mothers to their kids. Coat color and reproductive traits such as kidding interval were also mentioned as important traits by both goat keepers. Shorter kidding interval will increase flock size for marketing and replacement. It would be also helpful for genetic improvement program by increasing selection intensity though the improvement of kidding interval through selection may be slow because of the low heritability of the trait [6].

Mean ± SE values for dentition and some production and reproduction traits of does from own-flock ranking experiment are presented in Table 2. Body weight, number of kidding, twinning, number of kids born and number of kids weaned significantly (p < 0.001) influenced the ranking decision of both goat keepers. Dentition had significant (p < 0.001) effect only on Oromo goat keepers' trait preference of breeding does.

Comparing the mean values of does ranked as 1st best and poor quality, there were clear and logical differences in most of the attributes considered. This indicated that the farmers' choices of does were confirmed by the objective measurements. For instance, in the Arab goat population, the magnitude difference between the 1st best and inferior does in live weight, number of kids born and number of kids weaned were 2.22 kg, 1.83 and 1.70, respectively. Similarly, in the Oromo goat keepers, the difference between the two groups for the same traits were 2.66 kg of body weight, 1.62 numbers of kids born and 1.79 numbers of kids weaned. The longer ages of the best does indicate that keepers are willing to keep them for long service years to achieve their objectives. In the study areas, where performance and pedigree recordings are completely absent, proper recognition and application of the goat keepers' indigenous knowledge for selecting the best breeding does is possible option to start appropriate breeding programs. The picture of the two goat populations is depicted in Fig 1.

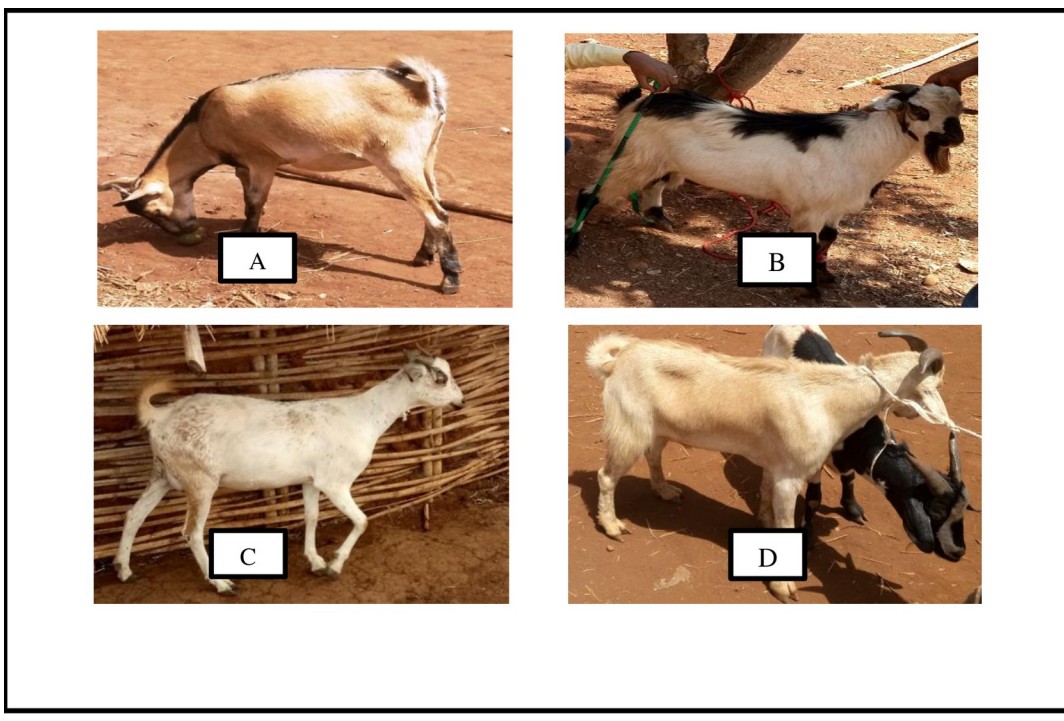

**Fig 1.** Representative pictures of adult Arab doe (A), young Arab buck (B), adult Oromo doe (C) and adult Oromo buck (D) included in the study.

### Doe traits in group-animal ranking experiments

Table 3 presents the lists of preferred doe traits in the group-animal ranking experiments. Coat color, body size, body conformation, body width and mothering ability, in that order, were found to be the most important traits in Arab goats, the sum of which accounted for 53.72% of the traits mentioned in Arab doe-ranking experiment. In Oromo doe-ranking experiment, about half (52.83%) of the mentioned traits were contributed by body size, twinning rate, body conformation, coat color and mothering ability.

### Buck traits in group-animal ranking experiments

Table 4 describes the lists of buck traits in group-animal ranking experiments. Coat color, body size, body conformation and body length were the four most important phenotypic traits which accounted for 54.12% and 52.38% of the traits mentioned by Arab and Oromo goat keepers, respectively, but with varying order. Coat color assumed the first priority with a magnitude of 18.04% followed by body size (17.53%), body conformation (9.79%) and body length (8.76%) in Arab goat keepers while the order of recurrence of traits in Oromo goat keepers was body size, coat color, body conformation and body length with magnitude of 20.63%, 17.99%, 7.41% and 6.35%, respectively.

### Comparisons of rankings with and without additional information of life history

Table 5 summarizes rank proportions before and following provision of information about life history of does and bucks. In Arab goat populations, of does that ranked as first, second, and third prior to provision of life history, 70.8%, 68.8% and 72.9% of them retained their position,

**Table 3. List of doe traits in group-animal ranking experiments.**

| | Traits | Arab | | Oromo | |
|---|---|---|---|---|---|
| | | Freq | % | Freq | % |
| 1. | Body size | 24 | 12.77 | 23 | 14.47 |
| 2. | Body conformation | 18 | 9.57 | 16 | 10.06 |
| 3. | Coat color | 28 | 14.89 | 15 | 9.43 |
| 4. | Color pattern | 10 | 5.32 | 11 | 6.92 |
| 5. | Body width | 16 | 8.51 | 5 | 3.14 |
| 6. | Body condition | 13 | 6.91 | 7 | 4.40 |
| 7. | Age | 13 | 6.91 | 9 | 5.66 |
| 8. | Horn length | 4 | 2.13 | - | - |
| 9. | Body length | 12 | 6.38 | 11 | 6.92 |
| 10. | Beauty/appearance | 7 | 3.72 | 12 | 7.55 |
| 11. | Ear size | 4 | 2.13 | - | - |
| 12. | Height | 6 | 3.19 | 10 | 6.29 |
| 13. | Mothering ability | 15 | 7.98 | 13 | 8.18 |
| 14. | Twinning rate | 13 | 6.91 | 17 | 10.69 |
| 15. | Kidding interval | 5 | 2.66 | 10 | 6.29 |
| | Sum | 188 | | 159 | |

respectively, after provision of life history. The corresponding values for Oromo goats were 52.1%, 47.9% and 50%. Unlike in does, the attached life history information only minimally altered respondents' decision in buck-group ranking. For example, in Arab goats, only 6.2% and 2.1% of the respondents changed their ranks from 1st to 2nd and 1st to 3rd, respectively. The corresponding values in Oromo goats were 4.2% and 0%. The likely reason for this is that keepers tended to judge and select female animals based on their reproductive performance

**Table 4. List of buck traits in group-animal ranking experiments.**

| Traits | | Arab | | Oromo | |
|---|---|---|---|---|---|
| | | Freq | % | Freq | % |
| 1. | Coat color | 35 | 18.04 | 34 | 17.99 |
| 2 | Color pattern | 10 | 5.15 | 10 | 5.29 |
| 3. | Body size | 34 | 17.53 | 39 | 20.63 |
| 4. | Body width | 15 | 7.73 | 9 | 4.76 |
| 5. | Age | 9 | 4.64 | 10 | 5.29 |
| 6. | Fast growth | 6 | 3.09 | 8 | 4.23 |
| 7. | Body condition | 12 | 6.19 | 9 | 4.76 |
| 8. | Horn shape | 5 | 2.58 | 11 | 5.82 |
| 9. | Appearance | 6 | 3.09 | 9 | 4.76 |
| 10. | Body conformation | 19 | 9.79 | 14 | 7.41 |
| 11. | Temperament | - | - | 6 | 3.17 |
| 12. | Body length | 17 | 8.76 | 12 | 6.35 |
| 13. | Height | 8 | 4.12 | - | - |
| 14. | Heat tolerance | 5 | 2.58 | - | - |
| 15. | Libido | 7 | 3.61 | - | - |
| 16. | Horn orientation | - | - | 8 | 4.23 |
| 17. | Horn size | 6 | 3.09 | 10 | 5.29 |
| | Sum | 194 | | 189 | |

**Table 5. Rank proportions before and after provision of information in group-ranking.**

| Population | RBLH | RALH Does | | | RBLH Buck | | |
|---|---|---|---|---|---|---|---|
| | | 1 | 2 | 3 | 1 | 2 | 3 |
| Arab | 1 | **34 (70.8%)** | 10 (20.8%) | 4 (8.3%) | **44 (91.7%)** | 3 (6.2%) | 1 (2.1%) |
| | 2 | 6 (12.5%) | **33 (68.8%)** | 9 (18.8%) | 2 (4.2%) | **44 (91.7%)** | 2 (4.2%) |
| | 3 | 6 (12.5%) | 7 (14.6%) | **35 (72.9%)** | 2 (4.2%) | 1 (2.1%) | **45 (93.8)** |
| Oromo | 1 | **25 (52.1%)** | 13 (27.1%) | 10 (20.8%) | **46 (95.8%)** | 2 (4.2%) | 0 (0%) |
| | 2 | 11 (22.9%) | **23 (47.9%)** | 14 (29.2%) | 2 (4.2%) | **45 (93.8%)** | 1 (2.1%) |
| | 3 | 11 (22.9%) | 13 (27.1%) | **24 (50%)** | 0 (0%) | 1 (2.1%) | **47 (97.9%)** |

RBLH = Rank before provision of life history; RALH = Rank after provision of life history; unchanged ranks are given along the diagonal.

and mothering ability apart from physical appearance (body size, coat color and body conformation) and dental examination. Similar research findings were reported in phenotypic group-animal ranking experiments for sheep [10] and goat breeds in Ethiopia [6] and for Ankole cattle in Uganda [28].

## Conclusions

Breeding objective traits were identified for Arab and Oromo goat populations through phenotypic ranking approaches (own-flock and group-animal ranking experiments) to design breeding programs. Given large number of traits identified in the present study, it would be useful to include only few priority traits in order to keep the breeding programs as simple as possible and for easy implementation under smallholders' circumstances. In the own-flock ranking experiments, keepers focus on productive, reproductive and behavioral traits whereas in the group-animal ranking experiments there was a generally tendency to focus on observable physical traits like coat color, body size and body conformation for both does and bucks. Thus, simultaneous use of both methods for identification of breeding objective traits in similar production systems is advisable.

## Supporting information

**S1 File. Raw data for breeding buck and doe phenotypic ranking.**
(XLSX)

**S2 File. R script for breeding buck and doe phenotypic ranking data analysis.**
(DOCX)

**S3 File. Formats used in the phenotypic ranking.**
(DOCX)

## Acknowledgments

The corresponding author gratefully acknowledges the Federal Ministry of Education, Ethiopia, for the PhD fellowship award, farmers who allowed their goats free for inventory purpose and all experts and development agents in the study areas for their cooperation during data collection.

## Author Contributions

**Conceptualization:** Oumer Sheriff, Kefyalew Alemayehu, Aynalem Haile.

**Data curation:** Oumer Sheriff.

**Formal analysis:** Oumer Sheriff.

**Funding acquisition:** Oumer Sheriff, Kefyalew Alemayehu.

**Investigation:** Oumer Sheriff.

**Methodology:** Oumer Sheriff.

**Project administration:** Oumer Sheriff.

**Resources:** Oumer Sheriff.

**Software:** Oumer Sheriff.

**Supervision:** Oumer Sheriff, Kefyalew Alemayehu, Aynalem Haile.

**Validation:** Oumer Sheriff, Kefyalew Alemayehu, Aynalem Haile.

**Visualization:** Oumer Sheriff, Kefyalew Alemayehu, Aynalem Haile.

**Writing – original draft:** Oumer Sheriff.

**Writing – review & editing:** Oumer Sheriff, Kefyalew Alemayehu, Aynalem Haile.

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
