## [Decision Letter · Decision Letter 0]

25 Feb 2021

PONE-D-20-38878

Phenotypic ranking experiments in identifying breeding objective traits of smallholder farmers in northwestern Ethiopia

PLOS ONE

Dear Dr. Sheriff,

Thank you for submitting your manuscript to PLOS ONE. After careful consideration, we feel that it has merit but does not fully meet PLOS ONE’s publication criteria as it currently stands. Therefore, we invite you to submit a revised version of the manuscript that addresses the points raised during the review process.

We look forward to receiving your revised manuscript.

Kind regards,

Dawit Tesfaye

Academic Editor

PLOS ONE

Journal Requirements:

2.We suggest you thoroughly copyedit your manuscript for language usage, spelling, and grammar. If you do not know anyone who can help you do this, you may wish to consider employing a professional scientific editing service.  

3.We note that you have indicated that data from this study are available upon request. PLOS only allows data to be available upon request if there are legal or ethical restrictions on sharing data publicly. For more information on unacceptable data access restrictions, please see http://journals.plos.org/plosone/s/data-availability#loc-unacceptable-data-access-restrictions.

5.Thank you for stating the following in the Funding Section of your manuscript:

"The financial support for this study was provided by Biotechnology Research Institute (BRI) of

Bahir Dar University, Ethiopia."

"The authors received no specific funding for this research "

Reviewers' comments:

Reviewer's Responses to Questions

**Comments to the Author**

1. Is the manuscript technically sound, and do the data support the conclusions?

Reviewer #1: Yes

Reviewer #2: Yes

2. Has the statistical analysis been performed appropriately and rigorously? 

Reviewer #1: No

Reviewer #2: Yes

3. Have the authors made all data underlying the findings in their manuscript fully available?

Reviewer #1: Yes

Reviewer #2: Yes

4. Is the manuscript presented in an intelligible fashion and written in standard English?

Reviewer #1: Yes

Reviewer #2: Yes

5. Review Comments to the Author

Reviewer #1: Umer et al. investigated goat phenotype ranking schemes for identification breeding objective traits of smallholder farmers in Northwestern Ethiopia. For this, the authors looked into Own-flock ranking scheme in which the farmers ranked does based on age, number of kidding, twinning ability, number of kids born per kidding and number of kids weaned and Group-animal ranking scheme in which breeding does and bucks were randomly selected and phenotype information was gathered and then the farmers were asked to rank the animals with and without phenotype information. Accordingly, the author indicated that body size, mothering ability, twinning rate, kidding interval and temperament are prioritized traits while in group-animal ranking scheme, traits including coat color, body size and body conformation were found to be the priority of farmers. All in all, this study is relevant and a step forward to identify relevant to establish goat breeding objectives that could fit to smallholder farmers in Northwestern Ethiopia. Nevertheless, before this article is accepted for publication, the following issues should be addressed.

1. Statistical testing in the majority of results provided in tables are missing

2. Tables 3, 4 and 5 are all described as Table 3-- please correct!

3. In table 2 - Please indicate possible significant differences between 1 and 2, 1 and 3 and /or 2 and 3 rakings using letters or any convenient symbols.

3. Please describe if there is any goat trait preference differences between the Oromo and Arab smallholder goat farmers.

4. In the conclusion part of this manuscript, the authors indicated that in the own-flock ranking s scheme farmers were interested in productive, reproductive and behavioral traits whereas in the group-animal ranking scheme, farmers were interested in coat color, body size and body. conformation. It will be interesting if the authors could address this controversy.

Reviewer #2: Authors carried out own-flock and group-animal ranking experiments on two goat breeds, the Arab and Oromo breeds, in northwestern Ethiopia. More than 15 productive, reproductive and behavioral traits were used to rank the animals according to the choice of the farmers. This is a very great effort in designing a breeding program at a community level. The paper is written very well. Authors need to state the result section as “result and discussion”. Moreover, supplementing a representative picture of the goats from each breed, with respect to the traits mentioned in Table 2 would enhance the manuscript.

6. PLOS authors have the option to publish the peer review history of their article (what does this mean?). If published, this will include your full peer review and any attached files.

Reviewer #1: No

Reviewer #2: No

---

## [Author Response · Author response to Decision Letter 0]

4 Mar 2021

Responses to the Editor’s and Reviewers' Comments

Dear editor and reviewers, we would like to thank you for giving us a chance to revise our manuscript. We are also very thankful for your thoughtful and thorough review of our manuscript. The comments are encouraging and you appear to share our judgment that the study and its results are important. Each comment has been carefully considered point by point and responded accordingly. Please see below, in blue, our response to your comments. All line numbers given in the authors’ response refer to the revised manuscript and the revision can be seen as track changes in the manuscript.

RESPONSE TO EDITOR’S COMMENTS

Authors’ response: Dear Editor, thank you so much for providing us the link. We have read it carefully and revised our manuscript according to the guideline.

Authors’ response: Dear, thanks for your concern. Indeed, we have tried all our best to improve the language usage, spelling, and grammar and make the manuscript clear to the readers.

Authors’ response: Dear Editor, thanks for your question. All relevant data are within the manuscript and its supporting information files.

Authors’ response: We thank you for your comment; we moved the ethics statement to the methods section of our manuscript.

5. Thank you for stating the following in the Funding Section of your manuscript:

"The financial support for this study was provided by Biotechnology Research Institute (BRI) of

Bahir Dar University, Ethiopia."

"The authors received no specific funding for this research". 

Authors’ response: Dear Editor, thanks for the comment. We removed the funding statement that appeared in the acknowledgement section of our manuscript. On the other hand, we included our amended statements within our cover letter.

RESPONSE TO REVIEWERS’ COMMENTS

Reviewer’s comment: 

Reviewer #1: Umer et al. investigated goat phenotype ranking schemes for identification breeding objective traits of smallholder farmers in Northwestern Ethiopia. For this, the authors looked into Own-flock ranking scheme in which the farmers ranked does based on age, number of kidding, twinning ability, number of kids born per kidding and number of kids weaned and Group-animal ranking scheme in which breeding does and bucks were randomly selected and phenotype information was gathered and then the farmers were asked to rank the animals with and without phenotype information. Accordingly, the author indicated that body size, mothering ability, twinning rate, kidding interval and temperament are prioritized traits while in group-animal ranking scheme, traits including coat color, body size and body conformation were found to be the priority of farmers. All in all, this study is relevant and a step forward to identify relevant to establish goat breeding objectives that could fit to smallholder farmers in Northwestern Ethiopia. Nevertheless, before this article is accepted for publication, the following issues should be addressed.

1. Statistical testing in the majority of results provided in tables are missing

Authors’ Response: Dear reviewer, thank you for your comment. We believe that statistical testing is needed and we included it in Table 2. The rest of the results in Tables 1, 3, 4 and 5 are qualitative data which, we believe, do not need any significance tests. 

2. Tables 3, 4 and 5 are all described as Table 3-- please correct!

Authors’ Response: We appreciate your comment. Sorry for the mistake we made. Now, we correct it and you may find the correction in the revised manuscript.

3. In table 2 - Please indicate possible significant differences between 1 and 2, 1 and 3 and /or 2 and 3 rakings using letters or any convenient symbols.

Authors’ Response: Dear reviewer, we appreciate you. Based on your suggestion, we indicate the possible significant differences between the 1st and 2nd, 1st and 3rd, 1st and inferior, 2nd and 3rd, 2nd and inferior, and 3rd and inferior rakings using letters.

4. Please describe if there is any goat trait preference differences between the Oromo and Arab smallholder goat farmers.

Authors’ Response: Dear reviewer, thank you so much for your comment. The goat trait preference difference between the Arab and Oromo goat keepers is not the objective of the present study. This issue is addressed in a previous study conducted by the same authors of the current study in 2019. For more information, you can refer an article entitled with “Production systems and breeding practices of Arab and Oromo goat keepers in northwestern Ethiopia: implications for community-based breeding programs” TROPICAL ANIMAL HEALTH AND PRODUCTION. 52, 1467–1478.

5. In the conclusion part of this manuscript, the authors indicated that in the own-flock ranking scheme farmers were interested in productive, reproductive and behavioral traits whereas in the group-animal ranking scheme, farmers were interested in coat color, body size and body conformation. It will be interesting if the authors could address this controversy.

Authors’ Response: Yes. In the conclusion part of our manuscript, we indicated that goat keepers in the own-flock ranking experiment were interested in productive, reproductive and behavioral traits whereas in the group-animal ranking experiment, they were interested in coat color, body size and body conformation. This could be due to the reason that keepers in the own-flock ranking experiment know the animals very well since they are the owners of the goats whereas in the group-animal ranking experiment, goat keepers rank animals brought from another area, hence they did not know them. Thus, this controversy was happened due to the knowledge of the goat keepers that they had on the ranked animals (ranking either their own goats or unknown goats). This issue is addressed well in the methods section of our manuscript (i.e., own-flock ranking experiment and group animal ranking experiment). As a result of this, we believe that it would be redundant if we address this same issue in the conclusion part.

Reviewer #2: Authors carried out own-flock and group-animal ranking experiments on two goat breeds, the Arab and Oromo breeds, in northwestern Ethiopia. More than 15 productive, reproductive and behavioral traits were used to rank the animals according to the choice of the farmers. This is a very great effort in designing a breeding program at a community level. The paper is written very well. Authors need to state the result section as “result and discussion”. Moreover, supplementing a representative picture of the goats from each breed, with respect to the traits mentioned in Table 2 would enhance the manuscript.

Authors’ Response: We are grateful for your comment. Now, we state the result section as “Result and discussion”. Furthermore, we depicted the pictures of the goat populations in Figure 1.

Best Regards,

Oumer Sheriff, on behalf of the authors

---

## [Editor Report · Decision Letter 1]

5 Mar 2021

Phenotypic ranking experiments in identifying breeding objective traits of smallholder farmers in northwestern Ethiopia

PONE-D-20-38878R1

Dear Dr. Sheriff,

We’re pleased to inform you that your manuscript has been judged scientifically suitable for publication and will be formally accepted for publication once it meets all outstanding technical requirements and comments raised by the academic editor. .

Kind regards,

Dawit Tesfaye

Academic Editor

PLOS ONE

Additional Editor Comments (optional):

The figure legend for figure 1 need to be revised as follows.

Figure 1. Representative pictures of adult Arab doe (A), young Arab buck (B), Adult Oromo doe (C) and Adult Oromo buck (D) included in the study.

Some of the pictures seem to be stretched to the left or right. Please correct it according to the instruction for authors.
---

## [Editor Report · Acceptance letter]

10 Mar 2021

PONE-D-20-38878R1 

Phenotypic ranking experiments in identifying breeding objective traits of smallholder farmers in northwestern Ethiopia 

Dear Dr. Sheriff:

I'm pleased to inform you that your manuscript has been deemed suitable for publication in PLOS ONE. Congratulations! Your manuscript is now with our production department. 

Kind regards, 

on behalf of

Dr. Dawit Tesfaye 

Academic Editor

PLOS ONE